# *Paramecium bursaria*—A Complex of Five Cryptic Species: Mitochondrial DNA *COI* Haplotype Variation and Biogeographic Distribution †

Magdalena Greczek-Stachura [1], Maria Rautian [2] and Sebastian Tarcz [3,*]

1. Institute of Biology, Pedagogical University of Krakow, Podchorążych 2, 30-084 Kraków, Poland; magdalena.greczek-stachura@up.krakow.pl
2. Laboratory of Protistology and Experimental Zoology, Faculty of Biology and Soil Science, St. Petersburg State University, Universitetskaya nab. 7/9, 199034 Saint Petersburg, Russia; mrautian@mail.ru
3. Institute of Systematics and Evolution of Animals, Polish Academy of Sciences, Sławkowska 17, 31-016 Kraków, Poland
* Correspondence: tarcz@isez.pan.krakow.pl
† urn:lsid:zoobank.org:pub:5E000D59-AC98-4CB5-903E-F0251A01B6A4.

**Abstract:** Ciliates are a diverse protistan group and many consist of cryptic species complexes whose members may be restricted to particular biogeographic locations. Mitochondrial genes, characterized by a high resolution for closely related species, were applied to identify new species and to distinguish closely related morphospecies. In the current study, we analyzed 132 sequences of *COI* mtDNA fragments obtained from *P. bursaria* species collected worldwide. The results allowed, for the first time, to generate a network of *COI* haplotypes and demonstrate the relationships between *P. bursaria* strains, as well as to confirm the existence of five reproductively isolated haplogroups. The *P. bursaria* haplogroups identified in the present study correspond to previously reported syngens (R1, R2, R3, R4, and R5), thus we decided to propose the following binominal names for each of them: *P. primabursaria*, *P. bibursaria*, *P. tribursaria*, *P. tetrabursaria*, and *P. pentabursaria*, respectively. The phylogeographic distribution of *P. bursaria* species showed that *P. primabursaria* and *P. bibursaria* were strictly Eurasian, except for two South Australian *P. bibursaria* strains. *P. tribursaria* was found mainly in Eastern Asia, in two stands in Europe and in North America. In turn, *P. tetrabursaria* was restricted to the USA territory, whereas *P. pentabursaria* was found in two European localities.

**Keywords:** ciliate protists; cryptic species; *Paramecium bursaria* complex; biogeography; *COI* haplotype variability

## 1. Introduction

The presence of cryptic species underlying conserved morphospecies (species identified based on shared morphology) has been found across ciliates. Many of them actually consist of cryptic species complexes. However, the issues concerning the biodiversity and biogeography of these species, as well as the determination of interspecific boundaries, remain one of the most important contemporary problems in protistology [1–4]. The possibilities of solving the above problems have increased significantly in recent decades, mainly due to the dynamic development of molecular biology techniques [5–9]. The main result of the application of molecular tools was the identification of a high genetic variation within individual species [10–12], which led to the description of many new species in recent years [7,13–19]. Moreover, the analysis of molecular variability in many cases was the only way to ascertain the existence of cryptic diversity within individual morphospecies [11,12,20]. In the genus *Paramecium* (Protista, Ciliophora, and Oligohymenophorea), new morphospecies like *P. ossipovi* [21], *P. buetschlii* [16], *P. grohmannae* [22], and *P. caudatum pakistanicus* [23], as well as cryptic species such as *P. primjenningsi*, *P. bijenningsi*, *P. trijenningsi* [24], and

*P. quindecaurelia* [25], have been recently described. Based on morphological features (morphological species concept), almost 20 morphospecies of the genus *Paramecium* have been classified into the following five subgenera: *Viridoparamecium*, *Chloroparamecium*, *Cypriostomum*, *Helianter*, and *Paramecium* [16,26,27]. Presumably, most of *Paramecium* morphospecies consist of cryptic species [24,28–31], which, due to their reproductive isolation, are biological species (biological species concept) [32]. Finally, the application of molecular delimitation mainly based on rDNA or mtDNA markers (phylogenetic species concept) allowed for effective verification of the above taxonomic hypotheses based on genetic distances and strong bootstrap support, which confirmed the existence of independent evolutionary lineages and made it possible to distinguish new taxa [16,23].

Determining the boundaries between individual ciliate taxa is difficult due to the complex species structure [33], the lack of samples from many ecosystems [34], and the proper selection of a DNA marker, especially when systematic identification in environmental DNA biodiversity surveys is based solely on molecular markers [35]. Despite the application of various DNA markers (ribosomal, mitochondrial, and nuclear), previous studies have shown that a fragment of cytochrome *c* oxidase subunit I-*COI* mtDNA should be the most promising tool for species identification in ciliates (including the genus *Paramecium*) [29,36–38].

*Paramecium bursaria* [39], studied in the current survey, is the only morphospecies in the subgenus *Chloroparamecium* and has a foot-shaped (similar to the imprint of a foot) cell of 80–150 μm with a compact type micronucleus. The species is relatively wide and short, being obliquely truncated or rounded anteriorly, broadly rounded posteriorly, and dorsoventrally compressed. Presumably, this species is distributed worldwide and is commonly found in ponds, streams, and other freshwater habitats [40,41]. A distinctive feature of this species is mutualistic endosymbiosis with some species of green algae [42,43]. Bomford [44] described the taxonomy of *P. bursaria* and documented six syngens (the term syngen has remained in use for *P. bursaria*) within this species. The syngens are reproductively isolated from each other. In cases where intersyngenic mating is observed, exconjugant cells die without dividing, so that reproductive isolation is maintained through F1 inviability [44,45]. Each *P. bursaria* syngen has a characteristic number of mating types, and the syngens have specific geographic distributions. Since the Bomford collection was lost, a new syngen numbering has been proposed based on a new representative collection of *P. bursaria* strains [29].

In the present study, we investigated 132 *COI* sequences obtained from *P. bursaria* strains classified into five syngens that were collected in remote geographic locations. *COI* sequences for 101 strains were obtained from the GenBank database, whereas 31 *COI* gene fragments were sequenced for the first time. On the basis of the obtained *COI* dataset, we were able to present phylogenetic relationships between *P. bursaria* syngens and to rename them by proposing binominal names as in other *Paramecium* cryptic species complexes [24,28]. Furthermore, this was the first attempt to analyze the distribution of *P. bursaria* with the use of haplotype networks to assess the biogeography of individual cryptic species of the complex.

## 2. Materials and Methods

### 2.1. Material

The origin of ciliate strains studied in the current work is presented in Figure 1 and Table S1 (Supplementary Material). Clonal cultures of all these strains were deposited in CCCS (Culture Collection of Ciliates and their Symbionts, Collection registered in WFCC, #1024) at the St. Petersburg State University.

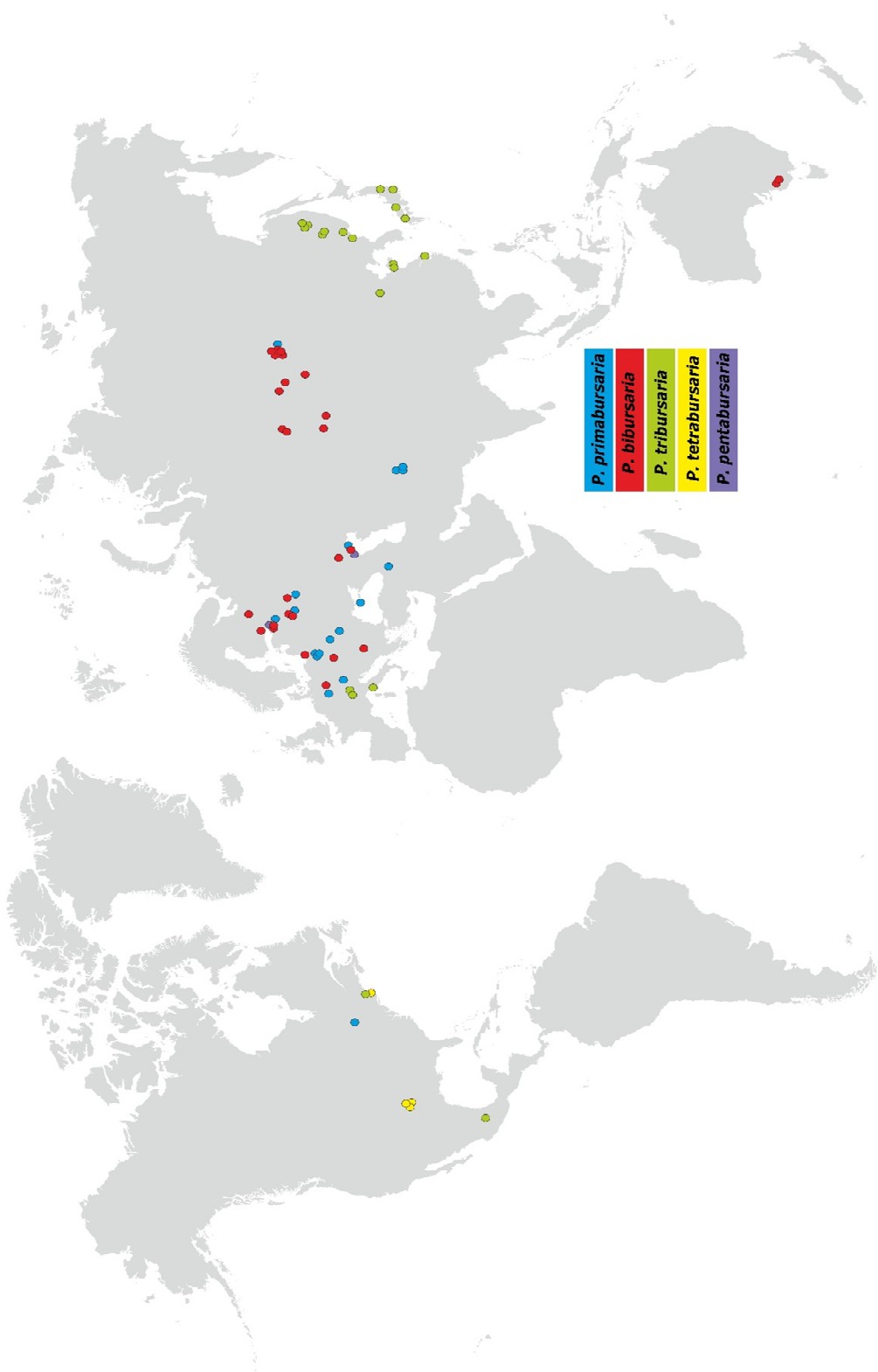

**Figure 1.** Geographic distribution of cryptic species of the *Paramecium bursaria* complex.

## 2.2. Methods

### 2.2.1. Culturing and Identification of *P. bursaria* Strains

Strains of *Paramecium bursaria* were obtained from the Culture Collection of Ciliates of St. Petersburg University. To prepare the cell culture for DNA isolation, paramecia were grown on lettuce medium inoculated with *Enterobacter aerogenes*. One volume of KK (Karakashian and Karakashian) buffer was added to two volumes of medium before feeding [46]. Ciliates were incubated at a temperature of 18 °C in the daily cycle of 12 light hours and 12 dark hours (12L/12D) in climate chambers (Angelantoni Life Science, Italy). The light intensity was 200 µmol m$^{-2}$ s$^{-1}$ and was maintained using a quantum sensor (model 189, Li-Cor, Inc., Lincoln, NE, USA). Syngen identification was performed by mating reactions of a strain of interest with standard strains representing all mating types of each syngen. The strain of interest was assigned to a certain syngen based on the strong clumping at the beginning of the mating reaction, observed mating couples, and survival of F$_1$ progeny. The origin of standard strains deserves special attention. All standard strains for syngens R1 and R2 were originally developed in the 1980s from a collection of stocks of unknown syngens using the round-robin mating test; strains of all eight mating types of syngen R1 were isolated from a population collected in Peterhof, a suburb of St. Petersburg; strains of all eight mating types of syngen R2 were isolated from a population collected in Lake Ladoga on Valaam Island. Four strains representing different mating types of syngen B1 were obtained from Prof. I. Miwa in the mid-1990s, and as it was the third syngen in the Russian collection, it was assigned the symbol "R3". Strains of different mating types in syngens R4 and R5 were identified based on the results of mating with standard strains of syngens R1–R3 and the results of round-robin mating tests. The set of standard strains is a matter of step-by-step substitutions, as *P. bursaria* strains lose their ability to actively mate with age. No microscopy characteristics was conducted.

### 2.2.2. Molecular Techniques

Genomic DNA of *Paramecium* was isolated (approximately 1000 cells were used for DNA extraction) from vegetative cells at the end of the exponential phase using the NucleoSpin Tissue Kit (Macherey-Nagel, Düren, Germany), according to the manufacturer's instructions for DNA isolation from human or animal tissue and cultured cells. The only modification was cell culture centrifugation for 20 min at 13,200 rpm. Then, the supernatant was removed and the remaining cells were resuspended in a lysis buffer and proteinase K. The proteinase K buffer step consisted of two parts: pre-lyse sample incubation at 56 °C for 3 h and lyse sample incubation at 70 °C for 10 min. The protocol details are available at https://www.mn-net.com/media/pdf/5b/d0/d9/Instruction-NucleoSpin-Tissue.pdf (accessed on 8 November 2021). Both the quantity and purity of the extracted DNA were evaluated using a NanoDrop-2000 spectrophotometer (Thermo Scientific, Waltham, MA, USA).

Fragments of the *COI* gene were amplified, sequenced, and analyzed. The *COI* fragment of mitochondrial DNA was amplified using a pair of primers, namely forward F388dT (5′-<u>TGTAAAACGACGGCCAGT</u>GGwkCbAAAGATGTwGC-3′) and reverse R1184dT (5′-<u>CAGGAAACAGCTATGAC</u>TAdACyTCAGGGTGACCrAAAAATCA-3′), and a protocol previously described in [36]. The amplification cycles were as follows: 4 min at 94 °C, followed by 5 cycles of 94 °C for 45 s, 45 °C for 75 s, and 72 °C for 90 s; 30 cycles of 94 °C for 45 s, 55 °C for 75 s, and 72 °C for 90 s; and a final extension at 72 °C for 8 min. PCR amplification was carried out in a final volume of 40 µL containing 30 ng DNA, 1.5 U Taq polymerase (EURx, Poland), 0.8 µL of 20 µM each primer, 10 X PCR buffer, and 0.8 µL of 10 mM dNTPs. In order to assess the quality of amplification, PCR products were electrophoresed in 1% agarose gel for 30 min at 85 V with a DNA molecular weight marker (MassRuler Low Range DNA Ladder, Thermo Fisher Scientific, Waltham, MA, USA).

To purify the PCR products, 5 µL of each product was mixed with 2 µL of Exo-BAP Mix (EURx, Gdańsk, Poland), and was subsequently incubated at 37 °C for 15 min, followed by another 15 min at 80 °C. Cycle sequencing was performed in both directions using BigDye Terminator v3.1 chemistry (Applied Biosystems, Waltham, MA, USA). The forward M13F

(5′-TGTAAAACGACGGCCAGT-3′) and reverse M13R (5′-CAGGAAACAGCTATGAC-3′) primers [36] were used for sequencing the *COI* fragment. Details of the sequencing procedure are derived from [30]. The studied *COI* sequences are available in the NCBI GenBank database (see Supplementary Table S1).

2.2.3. Data Analysis

Sequences were evaluated and chromatograms corrected using Chromas Lite v2.1.1 (Technelysium, Brisbane, Australia). Alignment of the studied *COI* mtDNA fragment was performed using BioEdit v7.2.5 software [47] and was checked manually. All sequences obtained were unambiguous and were used for further analyses. Mean uncorrected p-distances were calculated using Mega v6.0 [48]. Neighbor joining (NJ), maximum parsimony (MP), and maximum likelihood (ML) analyses were performed using Mega v6.0 program by bootstrapping with 1000 replicates. All of the positions containing gaps and missing data were eliminated. MP analysis was evaluated with the min–min heuristic parameter (at level 2) and bootstrapping with 1000 replicates. An HKY+G+I model for mtDNA (G = 0.758, I = 0.198) was identified as the best nucleotide substitution model for maximum likelihood tree reconstruction using Mega v6.0 software. Bayesian inference (BI) was performed using MrBayes v3.1.2 [49]; the analysis was run for 5,000,000 generations with the GTR+G+I model, and trees were sampled every 100 generations. All of the trees for the BI analysis were visualized using TreeView v1.6.6 [50].

The number of haplotypes (h), intraspecific haplotype diversity (Hd), and the nucleotide diversity ($\pi$) were determined with DnaSP v5.10.01 [51]. The haplotype network, representing the distribution and relationships among haplotypes of *Paramecium bursaria* strains, was reconstructed using the Median Joining method [52] implemented in PopART v1.7 software [53].

**3. Results**

*3.1. Paramecium bursaria Species Complex*

The current study revealed the existence of five distinct clades (Figures 2–4), which correspond to five *P. bursaria* syngens previously proposed by Greczek-Stachura et al. [29]. Therefore, to organize the nomenclature of *P. bursaria*-like taxa, confirmed by molecular methods and determined by strain crosses, as in other species complexes of the genus *Paramecium* (*P. aurelia* and *P. jenningsi*), we proposed the following nomenclature of syngens R1, R2, R3, R4, and R5: *Paramecium primabursaria* (former syngen R1), *Paramecium bibursaria* (former syngen R2), *Paramecium tribursaria* (former R3 syngen), *Paramecium tetrabursaria* (former syngen R4), and *Paramecium pentabursaria* (former syngen R5), respectively. When describing successive members of the *P. bursaria* complex, specific names should be created by adding a numerical prefix, similarly as for the *P. aurelia* complex [28] (for details see below Taxonomic summary, a subsection in the Discussion chapter).

*3.2. Analysis of COI mtDNA Sequences of the Paramecium bursaria Complex*

In the current survey, we studied a total of 132 sequences of *COI* mtDNA fragments obtained from *P. bursaria* strains collected in Europe, Asia, North America, and Australia (Figure 1 and Table S1). Additional GenBank records for other *Paramecium* subgenera and two *Tetrahymena* species (used as an outgroup) were included in phylogenetic *COI* tree reconstructions, thus, ultimately 215 mitochondrial DNA sequences were used (for details, see Table S2). After trimming, we used the mtDNA region containing cytochrome c oxidase subunit I (626–638 nt long depending on species) for the purpose of sequence comparison.

The intraspecific haplotype diversity (Hd) of the morphospecies *P. bursaria* was 0.966 for *COI* (*n* = 132), which indicates a significant differentiation within the *P. bursaria* species complex. Individual species were, however, characterized by a much lower intraspecific haplotype diversity (Hd) of 0.01 (Table 1). Within the studied *P. bursaria* complex, we identified 63 *COI* haplotypes, which were divided into individual cryptic species as follows: 9 haplotypes belonging to *P. primabursaria*, 18 haplotypes to *P. bibursaria*, 31 haplotypes to

*P. tribursaria*, 3 haplotypes to *P. tetrabursaria*, and 2 haplotypes to *P. pentabursaria* (Figure 3). Mutual relationships between haplotypes are presented later in the text. Details of DNA sequence variations among all of the studied cryptic species are shown in Table 1 and Table S3.

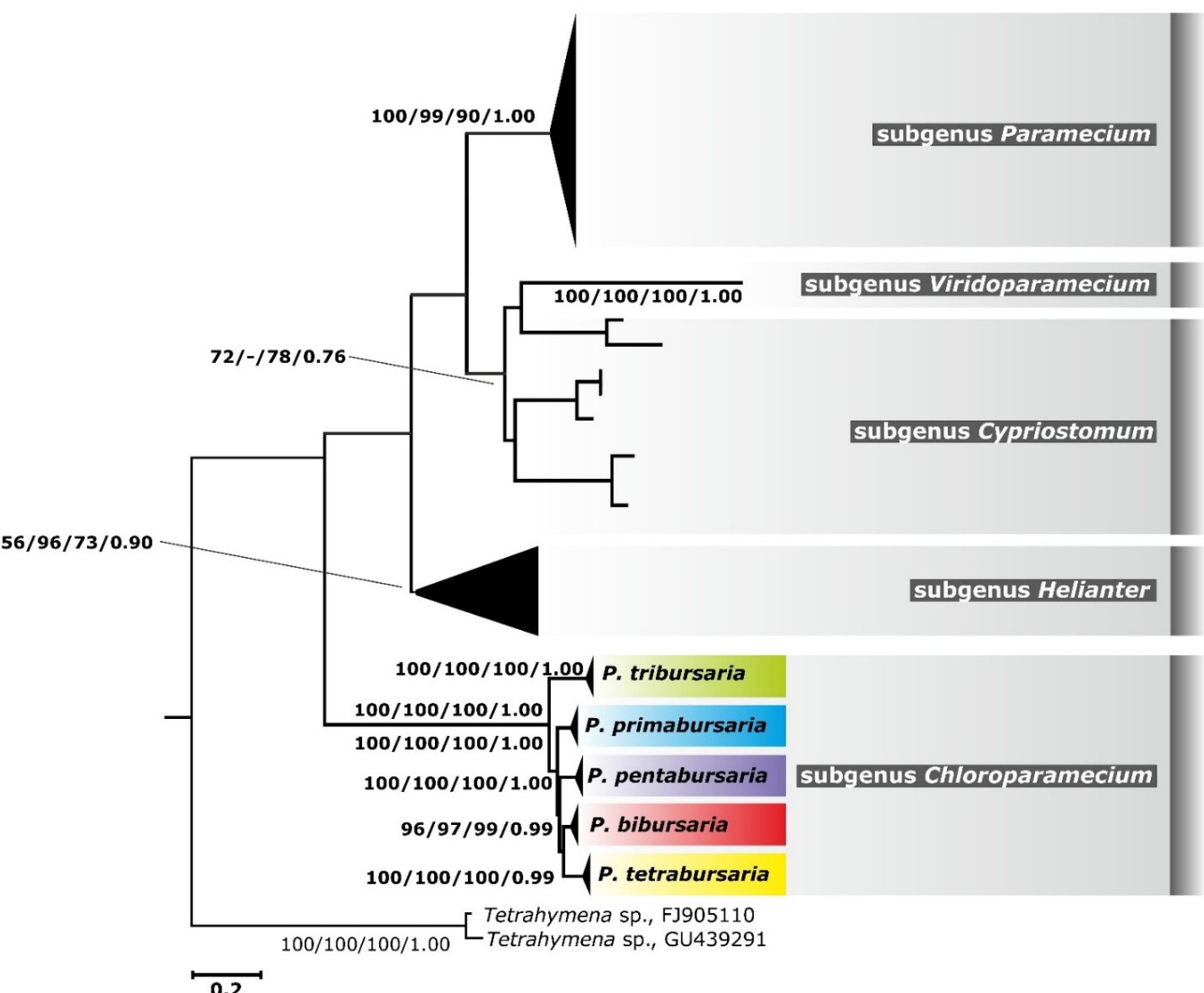

**Figure 2.** Phylogenetic tree constructed for 132 *Paramecium bursaria* strains and other *Paramecium* subgenera (two *Tetrahymena* species are used as an outgroup). The tree is built on the basis of the mitochondrial *COI* fragment using the Maximum Likelihood method (ML). Bootstrap values for neighbor joining (NJ), maximum parsimony (MP), maximum likelihood (ML), and posterior probabilities for Bayesian inference (BI) are presented. Bootstrap values lower than 50% (posterior probabilities < 0.50) are not shown. Dashes represent no bootstrap or a posterior value at a given node. All positions containing gaps and missing data are eliminated. Phylogenetic analyses are conducted using MEGA v6.0 (NJ/MP/ML) and MrBayes 3.1.2 (BI). The analysis involved 215 nucleotide sequences. There are 649 positions in the final dataset.

It is worth noting that the uncorrected p-distances within the *P. bursaria* complex are comparable to or even greater than those observed for species from the *P. aurelia* and *P. jenningsi* complexes (Table 1). The current study suggests that the *COI* mtDNA fragment is a good diagnostic tool for cryptic species identification in the *P. bursaria* species complex.

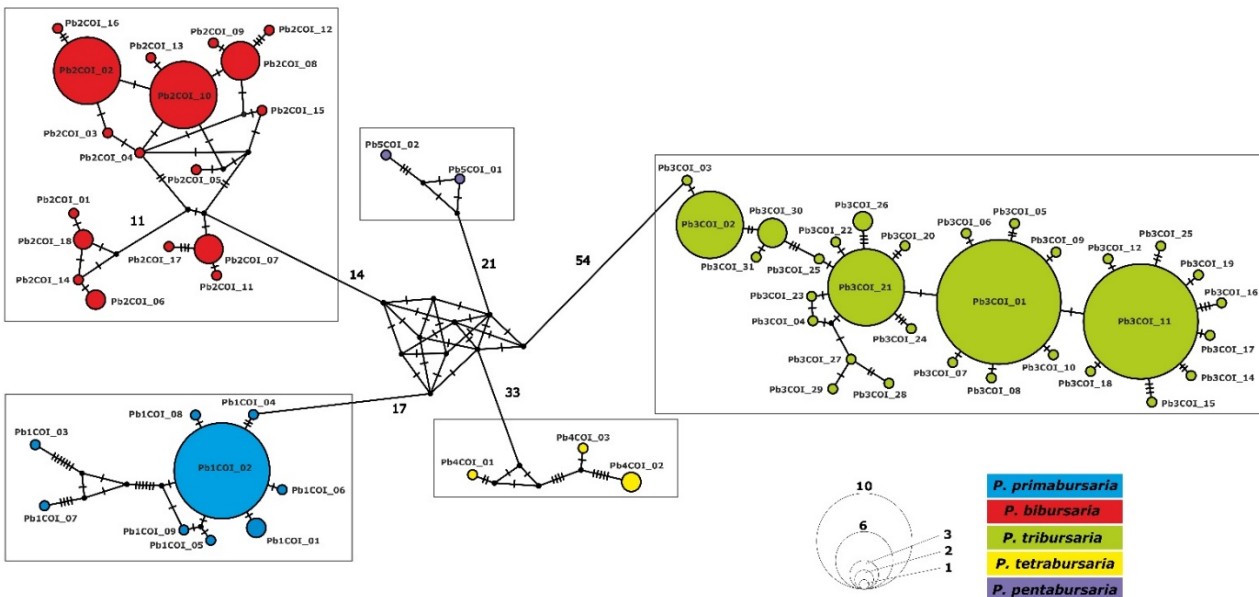

**Figure 3.** Haplotype network of the *Paramecium bursaria* complex constructed using 132 mitochondrial *COI* gene sequences. The network presents interrelationships among *P. bursaria* strains. Hatch marks on individual branches represent nucleotide substitutions (the corresponding number is provided for more than 10 substitutions). Analyses are conducted using the median joining method in PopART software v. 1.7.

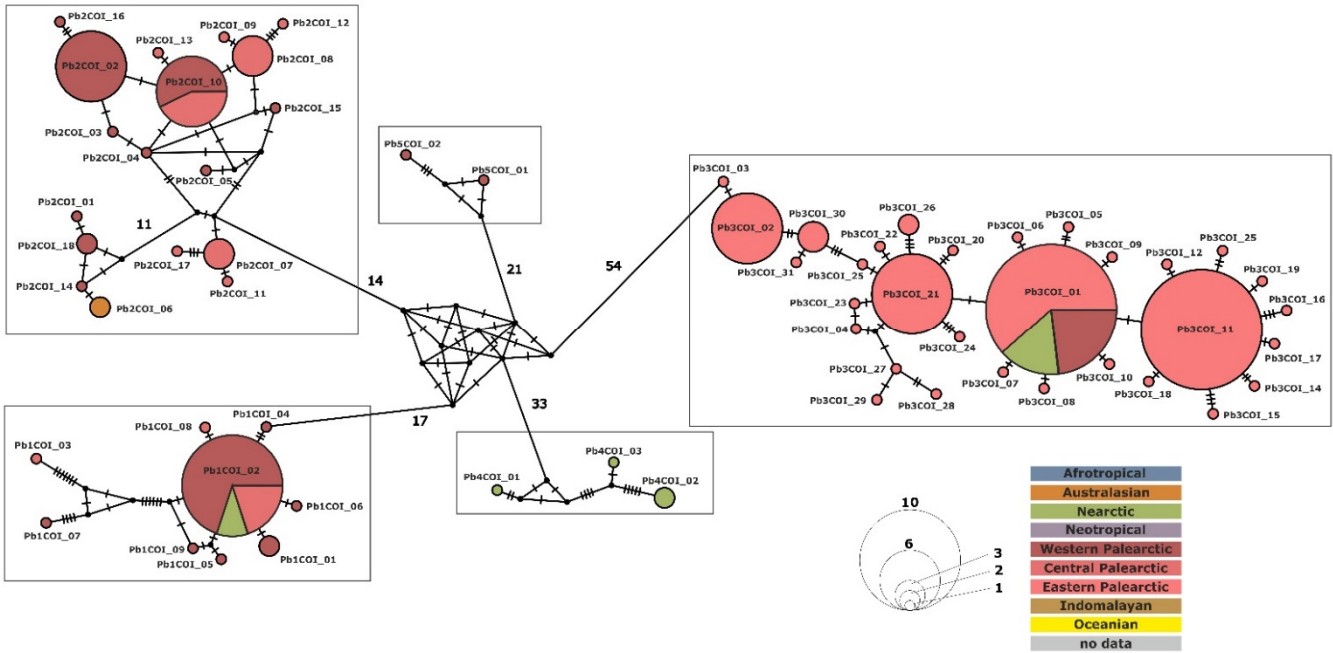

**Figure 4.** Haplotype network of the *Paramecium bursaria* complex constructed using 132 mitochondrial *COI* gene sequences. The network presents the origin and interrelationships among *P. bursaria* strains. Hatch marks on particular branches represent nucleotide substitutions (the corresponding number is provided for more than 10 substitutions). Analyses are conducted using the median joining method in PopART software v. 1.7.

**Table 1.** DNA sequence variability of species from the *Paramecium bursaria* complex in comparison with the *Paramecium aurelia* and *Paramecium jenningsi* species complexes.

| *Paramecium* Species | Number of Sequences N | Mean Uncorrected p-Distance | Number of Haplotypes h | Haplotype Diversity Hd | Nucleotide Diversity $\pi$ |
|---|---|---|---|---|---|
| *P. primabursaria* | 19 | 0.01 | 9 | 0.731 | 0.00552 |
| *P. bibursaria* | 37 | 0.01 | 18 | 0.920 | 0.0000029 |
| *P. tribursaria* | 70 | 0.01 | 31 | 0.918 | 0.00533 |
| *P. tetrabursaria* | 4 | 0.01 | 3 | 0.833 | 0.01133 |
| *P. pentabursaria* | 2 | 0.01 | 2 | 1.000 | 0.00618 |
| *P. bursaria* complex | 132 | 0.09 | 63 | 0.966 | 0.06351 |
| *P. jenningsi* complex * | 13 | 0.013 | 13 | 1.000 | 0.01779 |
| *P. aurelia* complex * | 15 | 0.084 | 15 | 1.000 | 0.07812 |

*-data obtained from [24].

### 3.3. Phylogenetic Relationship and Haplotype Analysis

Phylogenetic reconstruction based on 213 *COI* mtDNA fragments of all *Paramecium* species (Tables S1 and S2) revealed relationships among the 132 studied strains of the *P. bursaria* complex, as well as their position relative to other members of the genus *Paramecium*. All methods (NJ, MP, ML, and BI) of tree reconstruction showed a similar topology, thus we decided to present only the maximum likelihood phylogram (Figure 2) providing bootstrap/posterior probability values for the other methods used. The former *P. bursaria* clade (according to the present proposal of the *P. bursaria* species complex) was found to be distinct from other species of the genus *Paramecium*. In addition, it could be divided into two clades: one clade with the most distant *P. tribursaria* and the second clade containing all other species of the complex. It can be noticed that *P. bibursaria* and *P. tetrabursaria* are sister subclades in the second clade.

We constructed a haplotype network to examine interspecific relationships within the *P. bursaria* complex in more detail (Figures 3 and 4). The first *COI* network (Figure 3) presents mutual relationships among *P. bursaria* strains and indicates the existence of five well-separated haplogroups, corresponding to individual cryptic species. Generally, *P. bursaria*-like species are separated from each other by as many as 68–87 different nucleotides. Haplotypes within individual cryptic species differ from each other by 1 to 22 substitutions (Figure 3). The second network (Figure 4) presents the distribution of haplotypes in terms of the biogeography of the *P. bursaria* species complex.

### 3.4. Paramecium bursaria Complex Biogeography—Analysis of Haplotype Variability

As mentioned above, the analysis of intraspecific variation within the *P. bursaria* complex revealed the existence of 63 *COI* haplotypes (Figures 3 and 4), which could be divided into five haplogroups corresponding to five cryptic species: *P. primabursaria* (haplotypes Pb1COI_01-09), *P. bibursaria* (haplotypes Pb2COI_01-18), *P. tribursaria* (haplotypes Pb3COI_01-31), *P. tetrabursaria* (haplotypes Pb4COI_01-03), and *P. pentabursaria* (haplotypes Pb5COI_01-02). It should be noted that *P. primabursaria*, *P. bibursaria,* and *P. tribursaria* have one to four haplotypes that are characterized by a wide-range (sometimes intercontinental) distribution, whereas most haplotypes are usually restricted to one location (Figure 4). Overall, most of the studied haplotypes were identified as Palearctic (for better orientation, a division was introduced into the Western, Central, and Eastern Palearctic biogeographic ecozones), five were from the Nearctic biogeographic ecozone (Pb1COI_02, Pb3COI_01, Pb4COI_01-03) and one from the Australasian biogeographic ecozone (Pb2COI_06). An evaluation of the geographic distribution of the *P. bursaria* complex has shown that two cryptic species, *P. tetrabursaria* and *P. pentabursaria,* have a limited range, while the other species have a wider range of occurrence (Figure 1). *P. tribursaria* occurs mainly in Southeast Asia, plus two stands from Europe and two from North America.

*P. primabursaria* and *P. bibursaria* are found mainly in Europe and Central Asia. The only exception concerning *P. bibursaria* are two stands from Southern Australia. It is worth noting that *P. tetrabursaria* has so far been found only in North America (Nearctic biogeographic ecozone), while *P. pentabursaria* has been found in Western Russia (Palearctic biogeographic ecozone) (Table S1). Therefore, it can be assumed that some cryptic species of the *P. bursaria* complex differ to some extent in their ranges. Locations where two to three species of the *P. bursaria* complex were identified were rather rare (Figure 1): Astrakhan Nature Reserve or St. Petersburg and St. Petersburg surroundings were areas where *P. primabursaria*, *P. bibursaria*, and *P. pentabursaria* were found. However, ciliates (*P. pentabursaria*) in the St Petersburg Botanical Garden could have been introduced with plants from other locations.

## 4. Discussion

### 4.1. COI mtDNA as an Appropriate DNA Marker for Ciliate Species Delimitation

Almost 20 years ago, the idea of DNA barcoding (using the COI mtDNA gene fragment) was introduced into the scientific literature [54]. It was the first, widely known attempt to improve the taxonomic research. It quickly became apparent that there was no single universal DNA barcode for all living organisms [55]. Similarly, different markers have also been proposed for ciliates as the most appropriate tools for DNA barcoding [36,56,57]. For the genus *Paramecium*, two molecular markers are usually applied: different fragments of the nuclear rDNA [58–60] and the mitochondrial *COI* gene fragment [38,61]. However, it was demonstrated that in some cases, the more conserved rDNA fragments could lead to ambiguous results in taxonomic studies [38]. Mitochondrial genes generally evolve 5 to 10 times faster than nuclear genes [62] and are, therefore, more suitable molecular markers for closely related organisms, as has also been confirmed in ciliates belonging to the genus *Paramecium* [63]. Our previous study [29] indicated that only the application of *COI* allowed for distinguishing five syngens among the 26 studied *P. bursaria* strains. Based on the previous results, we decided to select this DNA fragment for the current comparative analysis of 132 sequences clustered into 63 haplotypes (Table S1).

### 4.2. Cryptic Species in the Genus Paramecium

Identifying species and determining their systematic position using molecular data are still interesting goals for evolutionary biologists. This work is particularly significant in closely related taxa, especially cryptic species, which are often indistinguishable based on morphological features [64]. The presence of such species has been reported in both multicellular [65,66] and unicellular organisms [67,68], including ciliates [69,70]. In the case of the latter, it is believed that almost 90% of the species have not yet been described [71]. Therefore, it is conceivable that the identification of cryptic species may significantly help in the assessment of ciliate biodiversity (as well as other microbial eukaryotes).

The presence of reproductively isolated groups, initially called syngens, was observed in the genus *Paramecium*, i.e., *P. aurelia* [45] and *P. bursaria* [44]. Later, syngens in *P. aurelia* were designated as sibling (cryptic) species based on isoenzyme pattern analysis [72,73] and strain crosses [28]. According to the biological species concept [32], cryptic species of the *P. aurelia* complex can be identified by mating reactions (conjugation) with standard strains [46]. Previous results, based on molecular analyses, have suggested that cryptic species are quite common in the genus *Paramecium*, not only in the *P. aurelia* and *P. jenningsi* complexes [24,74], but also in other morphospecies where the existence of cryptic species has not been formally described: *P. multimicronucleatum* [60], *P. putrinum* [30] and *P. calkinsi*, and *P. nephridiatum* [31]. Similarly, as in *P. bursaria*, molecular analyses of *COI* mtDNA fragments showed the existence of five clusters that corresponded to five syngens determined earlier based on the results of the mating reactions [29,75]. The current study, performed on 132 strains, confirmed the division into five clusters. Moreover, intraspecific analyses (Figures 3 and 4, and Table 1) have demonstrated that molecular genetic distances between individual clusters are similar to those between cryptic species clusters in the *P. aurelia* and *P. jenningsi* complexes [24]. Therefore, we suppose that it will be appropriate to give

binominal names to the five clearly distinct *P. bursaria* syngens and to simultaneously name the former *P. bursaria* [39], the Paramecium bursaria species complex (for details see below Taxonomic summary, a subsection in the Discussion section). The studied species meet the criteria of a species complex [64] because they can be differentiated based on strain crosses and molecular characteristics, but cannot be distinguished solely on the basis of their morphological features. It should be noted that each cryptic species is reproductively isolated and molecularly distant, and based on up-to-date sampling, seems to have a different geographic range [29].

*4.3. Paramecium bursaria Species Complex—Wide or Narrow Range Species?*

Microbial biogeography has been the subject of intense debate for several years [71,76]. There are two distribution models—the cosmopolitan model [2,77] and the moderate endemicity model [71,78]. According to Finlay's hypothesis, the high abundance of individuals observed within populations, short generation times, and high dispersal rates result in the absence of geographical barriers for particular species. In turn, the moderate endemicity model of ciliate biogeography indicates that two-thirds of protists may be distributed globally. However, some protist species are restricted to a specific region or area, i.e., they exhibit "local endemism" [78–80].

In the genus *Paramecium*, a narrow occurrence range is characteristic of *P. tredecaurelia*, *P. sonneborni*, or *P. quindecaurelia*, which are known from one to several locations [25,81,82], whereas *P. biaurelia* or *P. tertaurelia* are found worldwide [83]. Similarly, two distribution patterns are observed in the *P. jenningsi* complex, where *P. trijennngsi* has a wide range of occurrence, whereas *P. primjenningsi* and *P. bijenningsi* are restricted to two to three sampling sites [38]. Sonneborn [45] compared Paramecium species, considering features such as mating type inheritance, immaturity interval, maturity period, autogamy, selfing, and fission rate, as well as geographic distribution and proposed an inbreeding-outbreeding continuum. *P. bursaria*, which represents extreme outbreeders (many mating types, long periods of maturity and immaturity, low fission rate), is supposed to be a widespread taxon. Based on our list of sampling sites, we observed both narrow (*P. tetrabursaria* and *P. pentabursaria*) and wide range (*P. primabursaria*, *P. bibursaria*, and *P. tribursaria*) cryptic species. We are aware that the biogeographic division might be different, as the current analyses are limited to 56 locations (Table S1). Most of them were situated in the Western Palearctic biogeographic ecozone (27), followed by the Eastern Palearctic biogeographic ecozone (14), the Central Palearctic biogeographic ecozone (10), the Nearctic biogeographic ecozone (4), and finally the Australasian biogeographic ecozone (1). The presence of *P. bursaria* cryptic species in biogeographic ecozones such as Afrotropical, Neotropical, Indomalayan, and Oceanian zones has not been confirmed so far, which may be related to the relatively poorly sampled tropical areas. Interestingly, the only locality of the *P. bursaria* complex in the Australasian biogeographic ecozone is in the southern part of the continent, where the climate is close to moderate. Thus, up-to-date Paramecium species sampling has shown that the *P. bursaria* complex is widely distributed in moderate climates, and it has not been found in the tropics. This hypothesis may be supported by the fact that *P. bursaria* strains with endosymbiotic algae show higher growth rates at lower temperatures, while *P. bursaria* strains without algae exhibit higher growth rates at higher temperatures [84,85]. However, it is still likely that more intensive sampling in the tropical zone could change the biogeographic pattern of individual species of the *P. bursaria* complex. It can be assumed that temperature is one of the most important factors affecting the occurrence of species of the genus Paramecium. For example, cryptic species of the *P. jenningsi* complex were mainly found in the tropics [38]. The *P. aurelia* complex comprises cosmopolitan species that inhabit areas ranging from cold to tropical (*P. primaurelia*), non-tropical, widespread species (*P. biaurelia*), as well as species occurring in warm climates (*P. quadecaurelia* and *P. sonneborni*) [81,82].

Phylogenetic network analysis has shown that there are one to four dominant haplotypes in the case of *P. primabursaria*, *P. bibursaria*, and *P. tribursaria*. Some of them

show a wide, even intercontinental distribution, for example Pb1COI_02 or Pb3COI_01. The remaining haplotypes mainly have a narrow distribution restricted to one locality. The existence of similar haplotype network patterns (one to four dominant haplotypes and most haplotypes from a single locality) have been observed in *P. biaurelia* [86] or in *P. trijenningsi* [38]. This may indicate the absence of geographic barriers and the relatively rapid spread of closely related *P. bursaria* populations, with temperature being a key factor. Most importantly, the division into five haplogroups (Figures 3 and 4) does not correspond to the results obtained for the endosymbiotic algae inhabiting the studied species, i.e., the correlation between *P. bursaria* cryptic species and endosymbiont species, as well as between endosymbiont species and geographic distribution. It can therefore be assumed that the emergence of species within the *P. bursaria* complex was an earlier event than the association of individual ciliate species with endosymbiont species [42].

*4.4. Taxonomic Summary*

Class Oligohymenophorea de Puytorac et al., 1974 [87]
Order Peniculida Fauré-Fremiet (in Corliss), 1956 [88]
Family Parameciidae Dujardin, 1841 [89]
Genus *Paramecium* O.F. Müller, 1773 [90]
Subgenus *Chloroparamecium* Fokin et al., 2004 [26]
*Paramecium bursaria* Focke, 1836 [39]
The *Paramecium bursaria* species complex Greczek-Stachura et al., 2021 [42]
Desciption of the morphospecies *Paramecium bursaria*
Etymology. The name is derived from the Latin word *bursa* (purse), which refers to the shape of the cell.

Morphological Description (after literature data): *Paramecium bursaria* has a foot-shaped cell (similar to the imprint of a foot) with length of 140 μm and a width of about 60 μm. Cells are flat, with an obliquely truncated anterior end at the level of the wide oral groove and a broadly rounded posterior end. Length: 80–150 μm, width: 40–56 μm. One micronucleus, of compact type, relatively large (length: 25 μm, width: 11 μm). The single micronucleus belongs to the "compact b" type and its size is 14 μm in length and 7 μm in width. Two contractile vacuoles (CV) are located close to the dorsal surface in the endoplasm directly beneath the cortex. The CV are of the "canal fed" type, with long collecting vesicles and several excretory pores in each vacuole. The pores connect the CVs to the exterior environment and are located in the cortex; the number of the pores per CV in a species is a distinctive feature [91]. Caudal cilia are present, approximately 18 μm in length. The direction of rotation during swimming is counterclockwise. The cell is packed tightly with unicellular symbiont algae. The average number of symbiotic algae is 590 per cell and depends on light conditions. Phylogenetic analyses have revealed that *Paramecium bursaria* harbors endosymbionts representing different species [42]. It is the "green" *Paramecium*, which can be easily recognized by its color. *P. bursaria* exposed to photosynthetically active radiation (400–700 nm) exhibited the photoaccumulation phenomenon [92,93]. The photoreceptor of *P. bursaria* is a rhodopsin-like protein located on the anteroventral side, specifically within the oral groove of the cell [94]. It is the freshwater species. It is characterized by extreme outbreeding [22] and a wide geographic distribution. The morphospecies *P. bursaria* is divided into five syngens (species) with four to eight mating types for each syngen (species) [10]. The *P. bursaria* strain collection is located at the University of St. Petersburg. The current molecular identification of the species from the *P. bursaria* complex based on sequencing of the *COI* mtDNA fragment confirmed the existence of five species.

*Paramecium primabursaria* nov. spec.

Holotype: 87 MS-1.
Type locality: St Petersburg vicinity, Russia (59°52′ N 29°54′ E), a pond in park.
Distribution: widespread in Europe, several locations in Asia, including the Baikalarea (easternmost location).

Occurrence/number of mating types: eight mating types.
May correspond to Bomford's syngen B6.
Gene sequence: cytochrome *c* oxidase subunit I sequence of the holotype specimen. Has been deposited in GenBank under OK356526 accession number.

*Paramecium bibursaria* nov. spec.

Holotype: Ek.
Type locality: St Petersburg, Russia (59°58′ N 30°14′ E), a pond in a public garden.
Distribution: widespread in Europe, and central part of in Asia, two localities in South Australia.
Occurrence/number of mating types: eight mating types.
Demonstrate very good intersyngenic mating reactions with *P. tetrabursaria*, which suggests that it corresponds to Bomford's syngen B4.
Gene sequence: cytochrome *c* oxidase subunit I sequence of the holotype specimen. Has been deposited in GenBank under JF708911 accession number.

*Paramecium tribursaria* nov. spec.

Holotype: T316.
Type locality: Tsukuba, Japan (36°03′ N 140°07′ E) no habitat details.
Distribution: widespread in eastern Asia (Russia, Japan, China), three locations in southern Europe, and one in North America (Boston, MA, USA).
Occurrence/number of mating types: eight mating types.
It is still maintained in several laboratories in Japan, and these strains were once used for the identification of strains from the current collection. That makes it the only link to the lost Bomford's collection.
Corresponds to Bomford's syngen B1.
Gene sequence: cytochrome *c* oxidase subunit I sequence of the holotype specimen. Has been deposited in GenBank under JF708900 accession number.

*Paramecium tetrabursaria* nov. spec.

Holotype: AB2-32.
Type locality: Boston, USA (42°21′ N 71°04′ W), a pond in public garden.
Distribution: restricted to USA area.
Occurrence/number of mating types: six mating types.
Demonstrate very good intersyngenic mating reactions with *P. bibursaria* which suggests that it correspond to Bomford's syngens B2.
Gene sequence: cytochrome *c* oxidase subunit I sequence of the holotype specimen. Has been deposited in GenBank under JF708916 accession number.

*Paramecium pentabursaria* nov. spec.

Holotype: AZ20-1.
Type locality: Astrakhan Nature Reserve, Russia (45°53′ N 48°35′ E), backwaters in the delta of the Volga river.
Distribution: restricted to two location in Europe.
Occurrence/number of mating types: four mating types.
Corresponds to Bomford's syngen B3 or B5.
Gene sequence: cytochrome *c* oxidase subunit I sequence of the holotype specimen. Has been deposited in GenBank under JF708905 accession number.

The current publication has been registered at ZooBank under the number: urn:lsid: zoobank.org:pub:5E000D59-AC98-4CB5-903E-F0251A01B6A4.

## 5. Conclusions

Sequence analysis of the gene encoding the *COI* mtDNA fragment from *P. bursaria* strains revealed that the mean genetic distance within the *P. bursaria* complex was 0.966 and the p-distance was greater than those observed for species from the *P. aurelia* and *P. jenningsi* complexes. Tree reconstruction demonstrated that the clade of the *P. bursaria*

species complex is distinct from the other species of the genus Paramecium and is divided into two clades: the first, *P. tribursaria*, and the second containing the remaining species of the complex (*P. bibursaria* and *P. tetrabursaria* are sister subclades). The haplotype network has shown that cryptic species from the *P. bursaria* complex are separated from each other by as many as 68–87 different nucleotides. Haplotypes within individual cryptic species differ from each other by one to 22 substitutions. The *P. bursaria* complex, which represents extreme outbreeders, is believed to be a widespread taxon. It should be taken into account that the current research is based on 56 mainly Palearctic locations of *P. bursaria* species, and up-to-date *Paramecium* species sampling has not identified a *P. bursaria* complex in the tropics. Evaluation of the geographic distribution of members of the *P. bursaria* complex has revealed that two cryptic species, *P. tetrabursaria* and *P. pentabursaria*, have a limited range, while *P. primabursaria*, *P. bibursaria*, and *P. tribursaria* show a wider range of occurrence. The current study has confirmed that the *COI* mtDNA fragment is a good diagnostic tool for cryptic species identification in the *P. bursaria* species complex. Based on the results of the phylogenetic analyses, we decided to propose binominal names for each haplogroup of the morphospecies of *P. bursaria*: *P. primabursaria*, *P. bibursaria*, *P. tribursaria*, *P. tetrabursaria*, and *P. pentabursaria*.

**Supplementary Materials:** The following are available online at https://www.mdpi.com/article/10.3390/d13110589/s1, Table S1: *Paramecium bursaria* complex strains used in the current study [25,29,36,75,95,96], Table S2: *Paramecium* strains (except *P. bursaria*) used in the current study [16,36,59–61,83,95,97–103]. Two *Tetrahymena* strains were used as outgroup, Table S3: p-distance matrix of the studied *COI* mtDNA fragments.

**Author Contributions:** Conceptualization, M.G.-S. and S.T.; methodology, M.R. and S.T.; software, S.T.; formal analysis, S.T.; investigation, M.G.-S., M.R., and S.T.; resources, M.R.; data curation, S.T.; writing—original draft preparation, S.T.; writing—review and editing, M.G.-S. and S.T.; visualization, S.T.; supervision, M.G.-S. All authors have read and agreed to the published version of the manuscript.

**Funding:** This research was funded by the Pedagogical University of Krakow and the Institute of Systematics and Evolution of Animals, Polish Academy of Sciences.

**Institutional Review Board Statement:** Not applicable.

**Data Availability Statement:** Not applicable.

**Conflicts of Interest:** The authors declare no conflict of interest.

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
