# Peer review of "Paramecium bursaria—A Complex of Five Cryptic Species: Mitochondrial DNA COI Haplotype Variation and Biogeographic Distribution†"

_diversity, doi:10.3390/d13110589_

Round 1
Reviewer 1 Report
Ciliate comprises many cryptic species due to their complex genomic and morphological structure. Species delineation is important for taxonomists and evolutionary biologists. Therefore, Greczek-Stachura et al. in their article titled: Paramecium bursaria - a complex of five cryptic species. 2 Mitochondrial DNA COI haplotype variation 3 and biogeographic distribution, investigated the crypticity and the geographical distribution of the P. bursaria species using mitochondrial gene sequences. The manuscript is well organized. The methods and methodologies were adequate. However, I cannot see any morphological comparisons here; this is the only lacking in this study.
There are some minor things to be fixed
- Mega v6.0 - rewrite as MEGA v6.0
- Line 135 - Add PCR cycle information
- Line 157 - Which model did you use for BI and ML analyses? How did you get the model?
- Line 199 - Move this model information to M & M part.
Reviewer 2 Report
Dear Authors,
I was very happy to see a manuscript about the species problem in ciliates and cryptic species. I highly appreciate the huge sampling and find the paper very interesting. It is, indeed, a very important contribution to the field of ciliatology. Analyses were well conducted and data interpretation is fine. However, I see many issues that need to be fixed before publication. My criticism should not be found to be destructive but constructive and should significantly improve the paper.
Major Issues
- English
The quality of writing is far from being acceptable. There are too many grammar issues (mistakes are also in the titles of chapters). Although this is not the job of a reviewer, I made almost 200 corrections on 12 pages!!! In my opinion, it is unacceptable to submit a paper of such poor linguistic quality. Please do not take this as an English correction of your paper, it needs to be checked again by a professional. There are also semantic issues (please see below).
- Violation of the International Code of Zoological Nomenclature (ICZN)
Unfortunately, ICZN (1999) was seriously violated three times in the present paper. The violations are, in fact, so dramatic that all nomenclatural acts (i.e. establishment of new species) would be invalid in their present form!!!
- Zoobank registration number of the work has to be provided in the Manuscript. Otherwise, the establishment of new taxa is invalid, since Diversity is an electronic journal. Please see Recommendation 8A of the International Commission on Zoological Nomenclature 2012. Amendment of Articles 8, 9, 10, 21 and 78 of the International Code of Zoological Nomenclature to expand and refine methods of publication. Bull Zool Nomencl 69:161-169.
- Holotypes or syntypes of new species must be fixed. The accession numbers of type specimens, deposition of material, diagnosis (even molecular one, but it must be there), etymology, and type locality must be provided. Please see Article 72.3 of the ICZN (1999): “Name-bearing types must be fixed originally for nominal species-group taxa established after 1999. A proposal of a new nominal species-group taxon after 1999 (unless denoted by a new replacement name (nomen novum) [Arts. 16.4, 72.7]), must include the fixation of a holotype [Art. 16.4] (see Article 73.1) or syntypes [Art. 73.2]. In the case of syntypes, only those specimens expressly indicated by the author to be those upon which the new taxon was based are fixed as syntypes.”
- Type locality must be provided. Please see also Art. 76.1.1 of the ICZN (1999). If the type locality is not provided, the species can be neotypified in the future (Art. 75.3.1 of the ICZN 1999). I am sure you do not want that someone will neotypify your new species due to the lack of type locality in the original description…
Please see ‘Taxonomic account’ in the following paper for a template on how to establish validly new ciliate taxa: Rataj, M.; Vd'ačný, P. Cryptic host-driven speciation of mobilid ciliates epibiotic on freshwater planarians. Molecular Phylogenetics and Evolution 2021, 161, e107174, doi: https://doi.org/10.1016/j.ympev.2021.107174.
- Please provide a new chapter “Taxonomic account” or “Taxonomic summary”, following the aforementioned paper (Rataj and Vd'ačný 2021).
- The following new names are difficult to pronounce due to the lack of a thematic vowel. It is highly recommended to include the thematic vowel in the following names: prim -> primabursaria, tetr -> tetrabursaria, pent -> pentabursaria. Please do not forget to change the corrected names throughout the manuscript and in figures.
- It is nomenclatural and taxonomic nonsense to “erase” P. bursaria and replace it with the P. bursaria complex. Very likely one of your cryptic species corresponds to P. bursaria and should be neotypified. For details and reasons for neotypification, see Remarks for Trichodina steinii (p. 11 in Rataj and Vd'ačný 2021). If it is not possible to do it, you need to discuss it and explain why you cannot assign any syngen to the name-bearing taxon P. bursaria. However, the name P. bursaria is available and, most importantly, this taxon name is valid according to the ICZN (1999). You cannot ignore it, even if it does not meet the molecular criteria currently used.
- The term P. bursaria species is often used incorrectly. It is really nonsense to say that P. bursaria species contains 5 species. It can contain 5 subspecies, 5 taxa, or 5 distinct lineages but not 5 species. This formulation is simply imprecise and incorrect. The correct formulation would read ‘species of the P. bursaria complex’ or ‘P. bursaria complex’. According to the ICZN (1999), P. bursaria is the nominotypical species of the complex, which was completely ignored in the manuscript. The complex is often incorrectly treated as P. bursaria. However, these are two different things - species complex is something different than an individual species [and species is not a species complex]. Please check the usage of P. bursaria and the P. bursaria complex very carefully throughout the manuscript and correct it when appropriate.
- I absolutely miss a table showing the comparison of p-distances among species of the P. bursaria complex. I would say that these values should be explicitly mentioned in the text and gathered in a separate table. This would be very helpful for readers. For Tetrahymena, p-distances greater than 4% are considered to be interspecific. I would like to see such information for Paramecium in the manuscript.
Minor issues
- Molecular cloning
It was not mentioned and described in M & M. However, as you have used M13 sequencing primers, I guess you performed it. Please clarify and provide information about cloning.
- Abbreviations of ML, MP, and NJ should be introduced in M & M or in the figure legends. MP is even not mentioned when referring to support values (there are 4 numbers, but only 3 statistical methods are listed in the figure legend...)
- I do not know what is an ecozone and how were the distinct ecozones delimited. Which exact criteria were used to recognize ecozones? Is 'biogeographic zone' not a more appropriate term?

Reviewer 3 Report
The authors analyzed genetic relationships of the Paramecium bursaria complex using mitochondrial COI sequence. They found five distinct haplotypes, and as those correspond to previously reported syngens, they decided to introduce the new binominal names for each of these haplotypes. They also discuss the biogeographical patterns of this species complex, but more extended sampling is needed to test their hypotheses. Overall, this study is well-designed, conducted, and written without any significant shortcomings.
I have only several minor comments/suggestions:
L54: Would you please specify from what bootstrap values? Is it from some phylogenetic inference? If yes, from which one. You also cite the paper (L55) about secondary molecular structures. May secondary structures also be used for molecular species delimitation?
L100: Consider citing some sources about how to perform mating-type experiments for people without this background.
L132: Please, if possible, add a citation for the primers as you did for another set of primers in L143.
L148: Please, add the software version numbers where it is missing (in the section Data analysis). Also, unify the style of writing the software version specification - for example, Mega v6.0 or Mega 6.0, but please do not alternate it. Also, after you specified the software version in the methods, there is no need to repeat it in the following sections, for example, in L199. It is a subtle thing, but your text is very well-edited, and these small things may, even more, improve a reading experience.
L160: If possible, would you extend a bit this part about the haplotype statistics and network? What are those methods and what type of research or on what kind of data are used, the advantages of these methods, etc.
L233: Could you add some expected nucleotide distances for cryptical species that are known for this molecular marker in phylogenetically close taxa?
L268: Could you label each main haplotype by its name in Fig. 4?
L306: Please change taxons to taxa.
L332: Please correct: features.e to features.
L385: Some papers in the Diversity journal have a conclusion. Although this paper is brief and straightforward (which is good), consider adding the conclusion for researchers that want to grasp the study in the most condensed form.
Round 2
Reviewer 1 Report
The author revised the manuscript well and can be accepted.
Author Response
Response to Reviewer #1:
We are very grateful for accepting our article “Paramecium bursaria - a complex of five cryptic species. Mitochondrial DNA COI haplotype variation and biogeographic distribution” to be published in Diversity journal.
Reviewer 2 Report
Dear Authors,
I am very happy with the revision. Now, the manuscript looks much better! I have checked it very carefully again and have, unfortunately, still detected quite many minor issues (almost 75 comments). Please see the PDF attached.
I will repeat here only two more important issues:
1 . Please provide GPS coordinates for type localities.
2. Type species is an incorrect term. It has to be replaced by the term "holotype". Type species is a type of a genus, not of a species. Please see, the ICZN (1999).
All other comments are only minor.

Reviewer 3 Report
Dear Authors,
The manuscript was significantly improved after the revision, and I do not have any other suggestions to add.
Author Response
Response to Reviewer #3:
We are very grateful for accepting our article “Paramecium bursaria - a complex of five cryptic species. Mitochondrial DNA COI haplotype variation and biogeographic distribution” to be published in Diversity journal.
